# Dissecting the Genetic Basis of Yield Traits and Validation of a Novel Quantitative Trait Locus for Grain Width and Weight in Rice

**DOI:** 10.3390/plants13060770

**Published:** 2024-03-08

**Authors:** Man Yin, Xiaohong Tong, Jinyu Yang, Yichen Cheng, Panpan Zhou, Guan Li, Yifeng Wang, Jiezheng Ying

**Affiliations:** State Key Laboratory of Rice Biology and Breeding, China National Rice Research Institute, Hangzhou 311400, China; 13545663721@163.com (M.Y.); jinyu.yang@syngentagroup.cn (J.Y.); 13872209530@163.com (Y.C.); zhoupanpan202206@163.com (P.Z.); li@ricescience.org (G.L.); wangyifeng@caas.cn (Y.W.)

**Keywords:** rice, yield trait, grain width, grain weight, quantitative trait locus

## Abstract

Grain yield in rice is a complex trait and it is controlled by a number of quantitative trait loci (QTL). To dissect the genetic basis of rice yield, QTL analysis for nine yield traits was performed using an F_2_ population containing 190 plants, which was developed from a cross between Youyidao (YYD) and Sanfenhe (SFH), and each plant in the population evaluated with respect to nine yield traits. In this study, the correlations among the nine yield traits were analyzed. The grain yield per plant positively correlated with six yield traits, except for grain length and grain width, and showed the highest correlation coefficient of 0.98 with the number of filled grains per plant. A genetic map containing 133 DNA markers was constructed and it spanned 1831.7 cM throughout 12 chromosomes. A total of 36 QTLs for the yield traits were detected on nine chromosomes, except for the remaining chromosomes 5, 8, and 9. The phenotypic variation was explained by a single QTL that ranged from 6.19% to 36.01%. Furthermore, a major QTL for grain width and weight, *qGW2-1*, was confirmed to be newly identified and was narrowed down to a relatively smaller interval of about ~2.94-Mb. Collectively, we detected a total of 36 QTLs for yield traits and a major QTL, *qGW2-1*, was confirmed to control grain weight and width, which laid the foundation for further map-based cloning and molecular design breeding in rice.

## 1. Introduction

Adequate food is extremely vital for people all over the world, and it can satisfy people’s basic sense of security, achieve social stability, and ensure national security. The agricultural crisis, especially the grain price crisis from 2008 to 2011, triggered global unrest [1]. In 2017, approximately 11% of the world’s population suffered from hunger, and it is expected that by 2050, climate change will put an additional 77 million people at risk of hunger [2]. In 2019 alone, approximately 340 million children worldwide suffered from micronutrient deficiency, and 144 million children under the age of 5 had developmental delays (http://www.fao.org/worldfoodsituation/csdb/en/, accessed on 21 February 2024). In the face of natural disasters, including rising global temperatures and unstable climate events, cultivating crops with high yield potential is crucial for ensuring food security [3]. Rice is one of the most important cereal crops and serves as a stable food for more than half the world’s population [4]. Rice is grown in over 100 countries around the world [5], and increasing rice production can eliminate hunger and ensure food security. However, rice production faces many challenges. The increase in carbon dioxide concentration in the air will reduce the nutritional quality levels of B vitamins, proteins, iron, and zinc in rice [6]. In rice variety improvement, it is often found that one yield trait has improved while another yield trait has deteriorated, resulting in the phenomenon of trade-offs [7,8]. To enhance rice production, more attention is directed to deeply exploring the genetic mechanisms of rice yield.

As a complex agronomic trait, grain yield per plant (GYPP) is mainly determined by three key components: number of panicles per plant (NPPP), number of filled grains per panicle (NFGPP), and grain weight [9]. The three component traits and other related yield traits are typical quantitative traits and are controlled by quantitative trait loci (QTL). During the last 30 years, large numbers of QTLs for the yield traits have been identified by the approaches of classical QTL mapping and genome-wide association analysis (GWAS) with biparental populations and the natural populations, respectively. Biparental genetic populations have been widely used in the identification of QTLs for rice yield traits with PCR-based or genome-resequenced linkage maps. In comparison with biparental genetic populations, natural populations with a large scale of rice varieties contain more genetic diversity, and their corresponding GWAS can improve map resolution. However, rare alleles are difficult to identify in GWAS studies. Plentiful recombination events and allele differences play an important role in detecting the QTLs underlying rice yield traits. In recent decades, great advances in rice functional genomics have provided the foundation to increase rice production. The elucidation of the genetic mechanisms underlying each yield component will certainly benefit the improvement of rice variety.

The NPPP is closely related to tillering ability and is easily affected by environment, with low heritability and regulation by multiple genes [10]. Tillering is a fundamental trait in rice that ensures the number of panicles and sink size [11]. Rice tillers include primary, secondary, and tertiary tillers, of which the secondary and tertiary tillers in the late tillering stage contribute little to yield. Rice tillering involves two processes: one is the formation of the axillary meristem, and the other is the outgrowth of the tiller bud [12]. *MOC1*, *MOC2*, and *MOC3* are three vital genes related to rice tillering, and they affect the growth of rice tiller buds and the formation of axillary meristems, respectively. The *MOC1* plant mutant almost completely loses its tillering ability and only produces one main culm, while the wild-type plant has multiple tillers. *MOC1* encodes a putative GRAS family nuclear protein, which is mainly expressed in axillary buds and functions to initiate axillary bud formation and promote its outgrowth [13]. *MOC2* mutant plants show a significant decrease in tiller number, reduced growth rate, and dwarfing characteristics. The *MOC2* mutation leads to a deficiency in FBPase activity, resulting in an insufficient supply of sucrose and possibly ultimately inhibiting the growth of tillering buds [14]. *MOC3* mutant plants only have a monoculm and are sterile. Morphological and histological studies have shown that the disruption of tillering bud formation in *MOC3* leads to a monoculm phenotype in the plant [15]. Outgrowth of the rice tiller bud is mainly regulated by phytohormones. As one of the direct factors affecting rice yield, the NPPP has been widely studied in recent years. In a previous study, a set of rice populations were used to genotype 700,000 single nucleotide polymorphisms (SNPs). A total of 15 novel QTL loci related to rice tillering were identified by the GWAS. The DNA sequences of the above 15 novel QTL loci were analyzed using extreme tiller number phenotypic accessions, and a total of five candidate genes were identified [16].

The NFGPP, directly determining grain yield, is affected by the number of primary branches (NPB) and the number of secondary branches (NSB) [17]. There are extensive variations in the NFGPP and its correlation among different accessions. A large number of QTL mappings have been carried out by using biparental genetic populations [18,19,20]. The QTLs controlling the NFGPP are distributed on 12 chromosomes, with the most on chromosome 1, and more than 180 QTLs have been located by incomplete statistics (www.grarene.org, accessed on 1 April 2023). Abundant QTLs controlling the NFGPP and its directly related panicle traits have been identified by GWAS [21,22]. *Gn1a*, encoding the enzyme osckx2 that degrades cytokinin, was the first QTL to be cloned in rice to control the NFGPP. *Gn1a* delected leads to the accumulation of cytokinin and increases the NFGPP [23]. *GNP1* encodes an essential enzyme GA20 oxidase in the gibberellin synthesis pathway, which affects the NFGPP by affecting the number of secondary branches [24].

Grain weight is generally estimated by 1000-grain weight (TGW), which is mainly related to grain size and the filling process [25]. The heritability of grain size is relatively high, and so is affected by genetic factors, while the filling degree is easily influenced by the environment [26]. Grain size is composed of grain length (GL), grain width (GW), and grain thickness (GT), of which the first two are essential factors affecting yield [27,28]. Up to now, 21 QTLs controlling rice grain weight and grain size have been cloned and distributed on 8 chromosomes, except for chromosomes 1, 10, 11, and 12. Except for *GLW7* [28] and *GW5* [29] which were identified by GWAS, *OsLG3* [30] was identified by linkage analysis and association analysis, and the remaining eighteen QTLs were all identified by map-based cloning. Among these QTLs, 13 were found to mainly control GL and GW, including *GS2* [31] on chromosome 2; *GS3* [32], *SG3* [33], *OsLG3* [30], *qLG3* [34], *GL3.1* [35], *TGW3* [36] and *GSA1* [37] located on chromosome 3; *GL4* [38] on chromosome 4; *TGW6* [39], *GW6a* [40], and *GL6* [41] located on chromosome 6; and *GLW7* [28] on chromosome 7. *GL7* [42] on chromosome 7 and *GS9* [43] on chromosome 9 only control grain size, without affecting grain weight. The remaining six QTLs mainly control GW and TGW, including *GW2* [44] and *TGW2* [45] on chromosome 2; *GS5* [46] and *GW5* [29] on chromosome 5; and *GW6* [47] and *GW8* [48] on chromosomes 6 and 8, respectively.

In this study, the correlations among nine yield traits, GYPP, NPPP, number of total grains per plant (TNSP), NFGPP, number of grains per panicle (NSPP), seed-setting rate (SSR), GL, GW, and TGW, were analyzed, and a linkage map covering the whole genome was constructed through genotyping each plant with the selected polymorphic markers using a YYD/SFH F_2_ population. Subsequently, QTLs controlling yield traits were identified using the combined information of genotype and phenotype. Furthermore, a QTL for GW and TGW, *qGW2-1*, was validated and narrowed down to smaller marker intervals with a set of near-isogenic line (NIL) population and the populations derived from the residual heterozygotes.

## 2. Results

### 2.1. Phenotypic Variation

To dissect the genetic basis of rice yield, two indica varieties, Youyidao (YYD) and Sanfenhe (SFH), that showed significant differences inn yield traits were selected as parents to develop an F_2_ population (Figure 1a,b).

Nine yield traits were investigated in the parental varieties and the corresponding F_2_ population. The female parent YYD showed 284.6%, 512.2%, 611.6%, and 62.7% more NPPP, TNSP, NFGPP, and NSPP; as well as 17.54% and 264.8% higher SSR and GYPP than the male parent SFH. However, the grains of YYD were 31.7% shorter, 21.0% narrower, and 48.5% lighter than those of SFH (Figure 2).

In the F_2_ population, we found that wide variations existed in the phenotypic distribution of the nine yield traits (Figure 3). Significant transgressive segregations were observed in the distribution of the NFGPP, NSPP, SSR, and GYPP. All the traits showed continuous segregation, suggesting that they were controlled by multiple genes and were suitable for QTL analysis.

### 2.2. Correlations between Yield Traits

The correlation coefficients between yield traits are exhibited in Figure 4. GYPP positively correlated with all six of the yield traits except for the grain size traits. No correlation was observed between GYPP, GL, and GW. NPPP showed positive correlations with TNSP, NFGPP, GYPP, and negative correlations with NSPP, SSR, and GW. For the relationship among the four grain/spikelet number traits including TNSP, NFGPP, NSPP, and SSR, significantly positive correlations were observed among all the traits, except that TNSP showed a negative correlation with SSR. For the relationships among the three grain size/weight traits including GL, GW, and TGW, significantly positive correlations were found among them. Negative or no correlations were observed between the three grain size/weight traits and the four grain/spikelet number traits, except that GW showed a slight positive correlation with SSR.

### 2.3. Creation of a Linkage Map

Based on genotyping 190 individuals with 133 DNA markers (82 SSR, 51 InDel), a linkage map was constructed with 12 linkage groups corresponding to 12 rice chromosomes (Figure 5). The linkage maps spanned 1831.7 cM. The genetic distances between neighboring markers ranged from 0.5 to 36.4 cM, with an average interval of 15.1 cM.

### 2.4. QTLs for Rice Yield Traits

#### 2.4.1. QTLs for GYPP and NPPP

Using the YYD/SFH F_2_ population, three QTLs for GYPP with relatively small effects were identified on chromosome 3, and the LOD peak values ranged from 4.34 to 5.86 (Figure 5, Table 1). All the three QTLs explained 31.34% of the GYPP’s variations, with the additive effects for enhancing the GYPP contributed by the YYD allele. No major QTL for the GYPP were detected. Four QTLs controlling NPPP were mapped on chromosomes 3, 4, 10, and 12. These QTLs collectively accounted for 47.01% of the total phenotypic variation. One major QTL with a peak LOD value of 7.61, *qNPPP3*, was mapped on chromosome 3 and was responsible for 17.85% of the NPPP’s variation. The additive and dominance effects showed opposite directions. Although the male parent SFH only had 3 to 5 tillers, the SFH allele at *qNPPP3* increased the NPPP with a positive additive effect of 5.5 and showed a negative dominant effect of −4.1.

#### 2.4.2. QTLs for Grain Number Traits

Four grain number traits including TNSP, NFGPP, NSPP, and SSR were evaluated to perform QTL analysis in this study (Figure 5, Table 1). For TNSP, five minor QTLs were detected on chromosomes 1, 3, 7, 10, and 12, and accounted for 6.19% to 13.15% of the phenotypic variation. For NFGPP, four QTLs were identified on chromosomes 3, 6, 7, and 12, with the LOD peak values ranging from 3.20 to 6.19. Together, these QTLs were responsible for 43.77% of the total phenotypic variation. One major QTL located on chromosome 7, *qNFGPP7*, scored a peak LOD value of 6.19 and showed the additive effect of −179.1 with the YYD allele increasing the filled grain number. For NSPP, four QTLs were mapped on chromosomes 1, 3, 4, and 7, and showed peak LOD values ranging from 3.39 to 11.92. All four of the QTLs collectively explained 49.96% of the observed phenotypic variation. Noticeably, one major QTL for NSPP, *qNSPP7*, was co-located with *qNFGPP7* in the same marker interval RM22039-JD7015 and showed an additive effect of −31.0 and a dominant effect of −11.4. Similarly, the female parent YYD allele increased the spikelets. Five minor QTLs regulating the SSR were identified on chromosomes 1, 3, 6, 10, and 12, respectively. These QTLs together accounted for 48.46% of the SSR’s variation and scored peak LOD values from 3.35 to 5.00.

#### 2.4.3. QTLs for Grain Size and Weight

Three grain size and weight traits, including the GL, GW, and TGW, were measured to perform QTL analysis in the YYD/SFH F_2_ population (Figure 5, Table 1). For the GL, three QTLs were detected on chromosomes 3, 7, and 11, which collectively accounted for 57.78% of phenotypic variation. One major QTL with an LOD value of 9.40, *qGL3*, was detected on chromosome 3 and was responsible for 36.01% of the total phenotypic variation, with the increasing GL contributed by the SFH allele. For GW, five QTLs were identified on chromosomes 1, 2, 3, and 10, which explained 81.23% of the phenotypic variation. On chromosome 2, two major QTLs with LOD values of 9.02 and 9.85, *qGW2-1* and *qGW2-2*, were identified to be responsible for 23.58% and 25.89% of the GW’s variations. For TGW, three QTLs were detected on chromosomes 2, 3, and 7, which explained 11.51%, 20.67%, and 10.07% of the total phenotypic variation, respectively. One major QTL, *qTGW3*, was identified in the marker interval RM15029-RM15353. The SFH allele underlying *qTGW3* showed an additive effect and increased the TGW.

### 2.5. Validation of qGW2-1

Three RH-derived populations in F_7_ with sequential heterozygous segments covering the target interval JD2001-JD2029, named YM1, YM2, and YM3, were developed to confirm the genetic effect and location of *qGW2-1*. The GL, GW, and TGW were continuously distributed in the three F_7_ populations (Appendix A). Three segmental linkage maps were constructed for the YM1, YM2, and YM3 populations (Figure 6). Based on the combination of genotype and phenotype information of each individual in the three populations, *qGW2-1* for the GW and TGW were identified in each population. No significant QTL effects for the GL were found in the three populations. The enhancing alleles for the GW and TGW contributed to the positive additive effects and were derived from the male parent SFH in all the three populations (Table 2). The variations explained in the GW were 31.54%, 28.28%, and 45.79% in the YM1, YM2, and YM3 populations. Based on the above results, we concluded that *qGW2-1* was located at the common segregating interval YS2005-YS2006 of the three populations, and the two flanking cross-over intervals YS2004-YS2005 in YM2 and YS2006-YS2027 in YM1. Therefore, *qGW2-1* was delimitated to the marker interval YS2004-YS2027, that is, it corresponded to ~2.94-Mb region in the Nipponbare genome (Figure 6).

For further validation of *qGW2-1*, one set of NIL plants in F_8_ that were produced through selecting from the YM3 population were planted during the growth season in Hangzhou. The phenotypic values of GL, GW, and TGW showed continuous distributions (Figure 7a). Obvious differentiations in the GW and TGW between the YYD and SFH homozygous genotypes were observed in the NIL population (Figure 7a–d). The YYD and SFH homozygous lines were in the low- and high-value areas of the GW and TGW, which suggests that *qGW2-1* is responsible for GW and TGW (Figure 7a).

## 3. Discussion

### 3.1. QTLs Dissect the Genetic Basis of Rice Yield Traits

To dissect the genetic basis of rice yield traits, we designed an F_2_ population that was derived from a cross of YYD and SFH with contrasting yield traits. As we expected, wide variations in the nine yield traits were observed in the population. After genotyping, phenotyping, and QTL mapping, we created a linkage map covering the rice genome and identified a total of 36 QTLs for the nine yield traits. The identified QTLs were distributed throughout the whole genome, except for chromosomes 5, 8, and 9 and explained from 6.19% to 36.01% of the total phenotypic variation. Except for the GYPP, TNSP, and SSR, one or two major QTLs that explained > 15% of the phenotypic variation were identified for the remaining six yield traits. For each yield trait, the number of QTLs that were detected in the YYD/SFH F_2_ population ranged from three to five. Similar to the earlier studies, the smallest number of QTLs was observed for the most complex trait, the GYPP, and no QTL could explain > 15% of the phenotypic variation. Through QTL analysis, it was found that there are QTLs that control different traits within the same molecular marker interval. In the molecular marker interval RM22039-JD7015 on chromosome 7, there are QTLs that control the GYPP, NFGPP, TNSP, GL, and TGW, respectively. It is believed that there may be a single factor for multiple effects or closely linked QTLs within a molecular marker interval.

### 3.2. Construction of Genetic Linkage Maps Is Important to Identify QTLs

Genetic linkage maps are the basis of QTL mapping, and appropriate molecular marker density is the key to the accuracy of QTL mapping. In this study, we constructed a low-density PCR-based genetic map containing 133 markers, of which the distance between some marker intervals was large, which may have caused some minor QTLs not to be detected. In order to confirm the reliability of the results of this map, we compared the QTL of yield-related traits found in this study with the previous mapping results. In this study, *qTNSP12* for the TNSP was identified in similar regions to the previous *qnspp12.1* and *qfgn12.1* controlling the TNSP and NFGPP, respectively [49]. *qGYPP3* and *qNFGPP3* for the GYPP and NFGPP were located in the intervals consistent with *qGY_S-3-1* controlling the GYPP and *qFG_S-3-1* controlling the NFGPP, which were detected by a high-density genetic map, respectively [50]. In our study, among the 11 QTLs for grain size and grain weight, *qGL3*, *qGW2-2*, and *qTGW3* were found adjacent to the cloned genes *GSA1* [37], *GS2* [31], and *SG3* [33], respectively.

### 3.3. Complex Correlations among Rice Yield and Yield Traits

Complex correlations were observed among rice yield and yield traits. Rice yield could be enhanced via the improvement of yield traits since the GYPP positively correlated with the yield traits (Figure 4). To improve the efficiency of rice breeding, elucidation of the relationship between yield and yield traits is very important and has been investigated in many studies [51,52]. Similar to our result, the GYPP was also found to positively correlate with the NPPP, NSPP, NFGPP, TGW, and SSR in the Pusa1266/Jaya recombinant inbred line population [53]. In the rice varieties released in China from 1978 to 2017, rice yield positively correlated with the NFGPP, TGW, NSPP, and SSR in the indica ecotype, whereas it positively correlated with panicle number per unit area in the japonica ecotype [26]. Previous studies have found that the colocation of QTLs responsible for different traits resulted in complex correlations [54,55,56]. In the present study, eight marker intervals each had at least two QTLs. All three of the QTLs for the GYPP, *qGYPP3*, *qGYPP6*, and *qGYPP7*, were all co-localized in the same genomic regions with the three QTLs for the NFGPP, *qNFGPP3*, *qNFGPP6*, and *qNFGPP7*, and their additive effects acted in the same direction, which might have resulted in the significant positive correlation between the GYPP and NFGPP, with the highest coefficient being 0.98.

### 3.4. qGW2-1 Is Confirmed to Be a Novel QTL for Grain Width and Weight

The genetic background of the population used for the initial mapping is generally complex, which may cause certain deviations in QTL interval. Before conducting QTL fine mapping and marker-assisted selection in variety breeding, it is necessary to verify the genetic effects and determine the region of the target QTL. In this study, one major QTL for GW and TGW, *qGW2-1*, was detected on chromosome 2, and was further validated and delimited to the target interval YS2004-YS2027 (~2.94 Mb). For grain size and weight, three major QTLs, including *GW2*, *GS2* and *TGW2*, were map-based cloned, and many minor QTLs were detected on chromosome 2. *GW2* negatively regulates GW and encodes a RING-type protein with E3 ubiquitin ligase activity [44]. *TGW2*, a semi-dominant QTL for GW and TGW, was mapped on the bottom of chromosome 2 and encodes CELL NUMBER REGULATOR 1 [45]. Another semi-dominant QTL, *GS2*/*OsGRF4*, encodes Growth-Regulation Factor 4 and increases GL, GW, and TGW through elevating its expression [31]. Although *GW2*, *GS2,* and *TGW2* are all located on chromosome 2, their locations are completely different from *qGW2-1*. Furthermore, many QTLs controlling grain size and weight are mapped on chromosome 2 and most of them are collected in the Gramene and Q-TARO databases (http://qtaro.abr.affrc.go.jp and http://qtaro.abr.affrc.go.jp, accessed on 8 April 2023). After comparing the QTL locations, no QTL for GW and TGW showed any overlapping interval with *qGW2-1*. Therefore, *qGW2-1* which was detected in this study is a novel QTL for GW and TGW. 

## 4. Materials and Methods

### 4.1. Plant Materials and Field Trials

Two indica varieties that show very significant differences in rice yield traits, including NPPP, NGPP, NSPP, SSR, GYPP, grain size and weight, YYD, and SFH, were selected to construct an F_2_ mapping population containing 190 individuals (Figure 1a). Three plants with heterozygous chromosomal segments covering *qGW2-1* were selected and self-crossed three times to produce three F_7_ populations by marker screening.

The field trials were carried out in two locations. We planted 190 plants of the F_2_ population and the two parents in Lingshui (110.0° E, 18.5° N) in Hainan Province, China. Three F_7_ populations and the parents, with each population containing of 190 plants, were grown in a paddy field in Hangzhou (120.2° E, 30.3° N) in Zhejiang Province, China. All the rice materials in this study were planted with a planting density of 16.7 cm × 26.5 cm during the rice-growing season. All kinds of field management were performed as in normal agriculture practice.

### 4.2. Phenotypic Measurement

After maturity, the panicles of each plant in the populations were harvested, respectively. The NPPP was counted manually. All the grains were threshed from the panicle, and the unfilled grains were separated from the filled grains and calculated artificiality. The filled grains were sun-dried and stored at room temperature for about three months. Then, the NFGPP and GYPP were measured by SC-A seed counting and grain weighting device (Wanshen Ltd., Hangzhou, China). Fully filled grains were separated by 3.5 mol/L NaCl solution and dried to measure the TGW, GL, and GW following the procedure reported by Zhang et al. [57]. Finally, the NGPP, NSPP, NSP, SSR, and GYPP were calculated.

### 4.3. DNA Extraction and DNA Marker Analysis

Total DNA was extracted from the young leaves of each plant in the populations according to the method of Zheng et al. [58]. PCR amplification was performed, and the products were visualized with 2.5% agarose gels with GelRed staining (Biotium, Fremont, CA, USA). The DNA markers used in this study included SSR markers and InDel markers. The SSR markers with the prefixes “RM” were selected from the public database (www.gramene.org, accessed on 9 April 2023). The InDel markers with the prefixes “JD” were taken from our previous study [36], and the InDel markers with the prefixes “YS” were newly designed using the online tool of Primer3.0 based on the 30× genome re-sequence of the parents YYD and SFH (Appendix A).

### 4.4. Construction of the Linkage Map

Based on the genotyping of the 190 plants of the F_2_ population with 133 SSR and InDel markers, twelve linkage groups corresponding to 12 chromosomes were established by using the computer program MAP-MAKER/EXP 3.0 [59]. The recombination frequencies among the markers in the same linkage group were converted into genetic distances using the kosambi function. An LOD score of 3.0 was used to determine the order of markers and the linkage group.

### 4.5. QTL Analysis and Statistical Analysis

Composite interval mapping using the Windows QTL Cartographer 2.5 was conducted to detect the QTLs for rice yield traits and further validate *qGW2-1* [60]. QTL analysis was carried out with 1000 permutations at the 0.05 probability level. An LOD score of 3.0 was fixed as the thresh value based on the 500 permutation tests for each trait. The correlation coefficients among different traits in the F_2_ population were evaluated by using Correlation Plot in the Origin2022 software.

## 5. Conclusions

The correlations among nine yield traits were analyzed using an F_2_ population derived from a cross of YYD and SFH with significant phenotypic differences. The GYPP positively correlated with six yield traits, except for GL and GW, and showed the highest correlation coefficient of 0.98 with the NFGPP. A genetic map was constructed and spanned 1831.7 cM throughout 12 chromosomes, with an average interval of 15.1 cM. A total of 36 QTLs for yield traits were detected on nine chromosomes, except for the remaining chromosomes 5, 8, and 9. The phenotypic variation that could be explained by a single QTL ranged from 6.19% to 36.01%. Furthermore, a major QTL, *qGW2-1*, was confirmed to control the GW and was narrowed down to relatively smaller intervals, defined in the marker interval of about ~2.94-Mb using three RH-derived populations in F_7_ and one set of NIL plants in F_8_. These results lay the foundation for further map-based cloning and molecular design breeding in rice.

## Figures and Tables

**Figure 1 plants-13-00770-f001:**
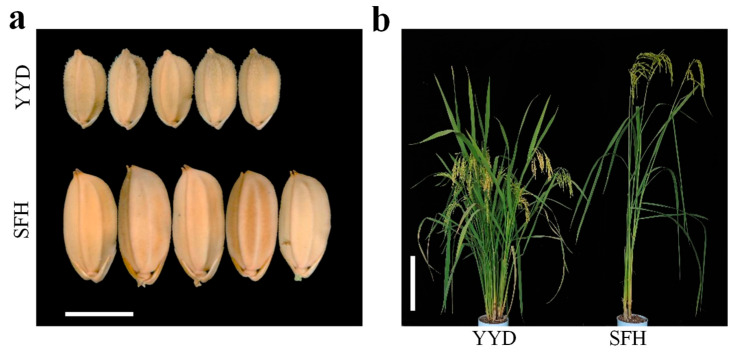
Comparison of grain size and plant type characters of YYD and SFH. (**a**) Grain phenotypes of two rice parents, YYD and SFH, bar: 5.00 mm. (**b**) Plant phenotypes of two rice parents, YYD and SFH, bar: 25 cm.

**Figure 2 plants-13-00770-f002:**
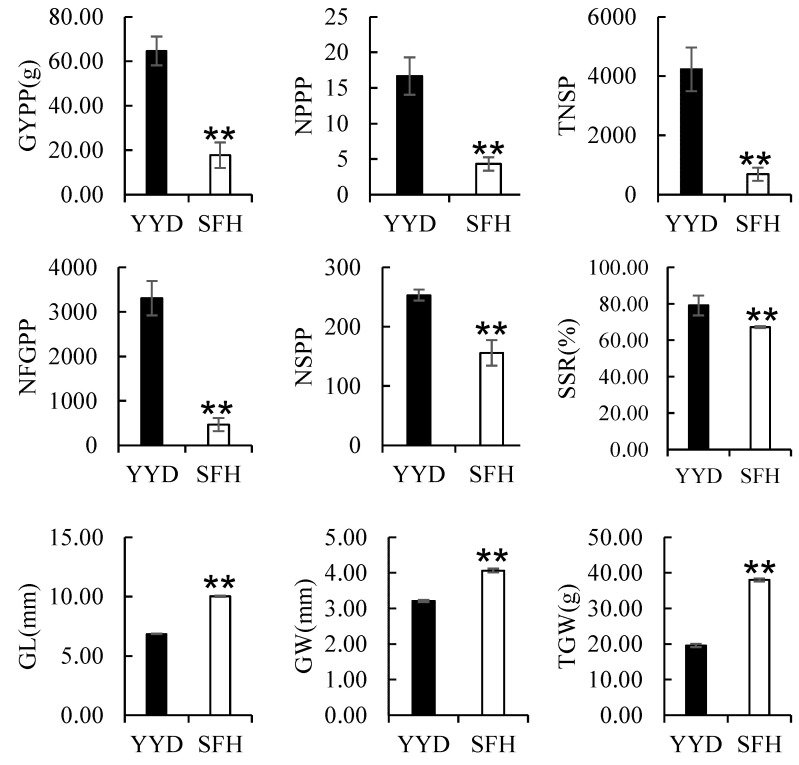
Comparison of 9 yield traits of YYD and SFH. Significant α = 0.01 **.

**Figure 3 plants-13-00770-f003:**
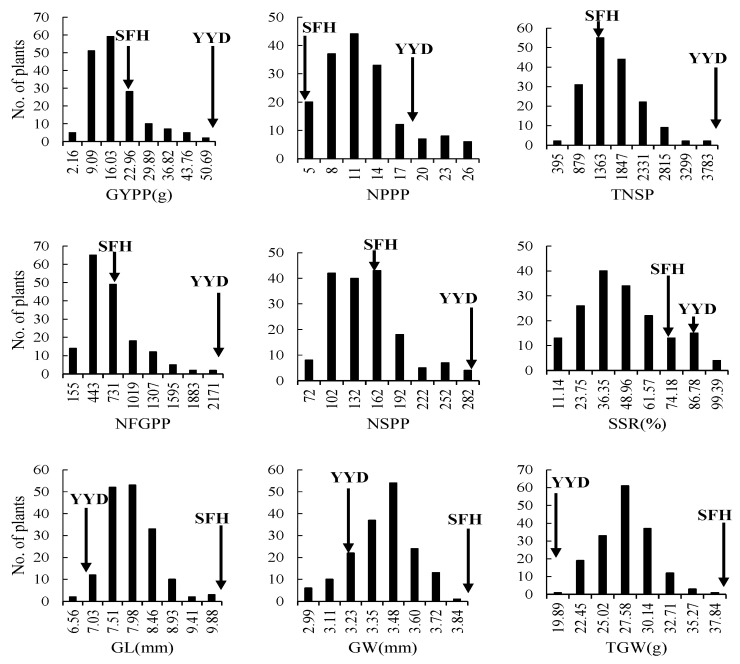
Frequency distribution of yield trait in the YYD/SFH F_2_ population. GYPP, grain yield per plant; NPPP, number of panicles per plant; TNSP, total number of spikelets per plant; NFGPP, number of filled grains per plant; NSPP, number of spikelets per panicle; SSR, seed-setting rate; GL, grain length; GW, grain width, and TGW, 1000-grain weight. Trait values of the parental materials are indicated by black arrows.

**Figure 4 plants-13-00770-f004:**
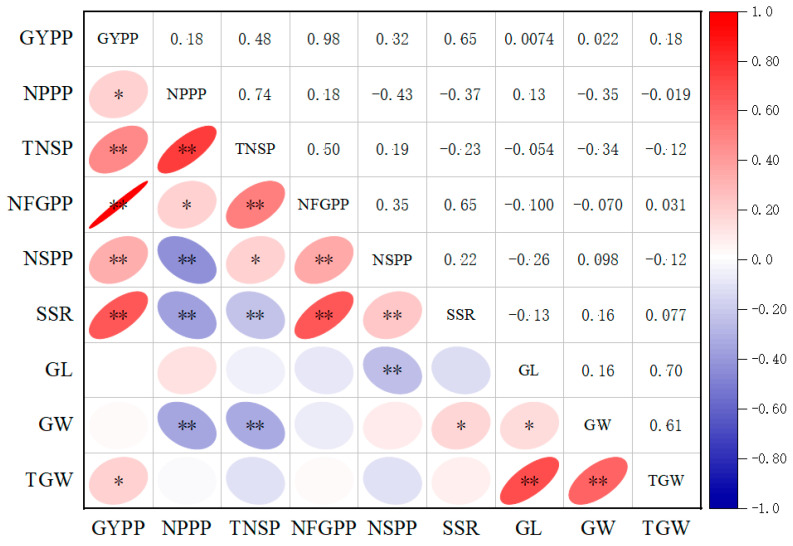
Correlation coefficients among 9 traits in the F_2_ population. The upper panel contains correlation coefficients, and the lower panel contains significance analysis. * and ** represent significance levels at 0.05 and 0.01, respectively.

**Figure 5 plants-13-00770-f005:**
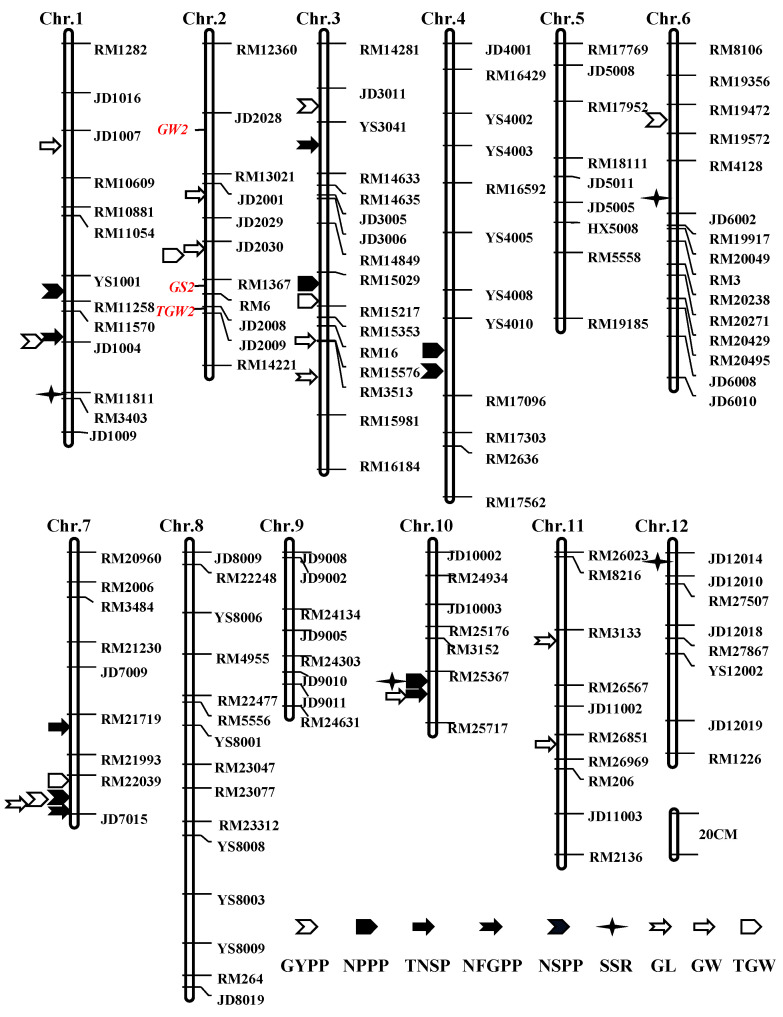
Genetic linkage map showing QTL positions detected in the F_2_ population. GYPP, grain yield per plant; NPPP, number of panicles per plant; TNSP, total number of spikelets per plant; NFGPP, number of filled grains per plant; NSPP, number of spikelets per panicle; SSR, seed-setting rate; GL, grain length; GW, grain width, and TGW, 1000-grain weight.

**Figure 6 plants-13-00770-f006:**
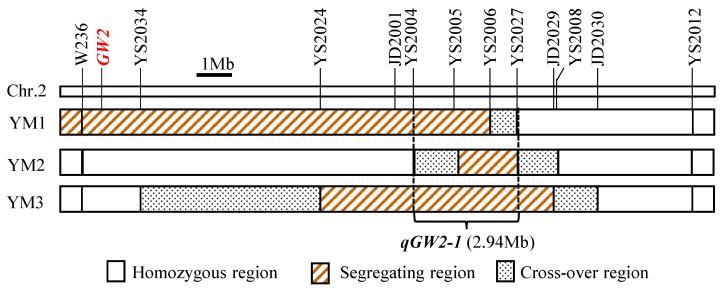
Genotype compositions of three F_2_ populations. Bar: 1 Mb.

**Figure 7 plants-13-00770-f007:**
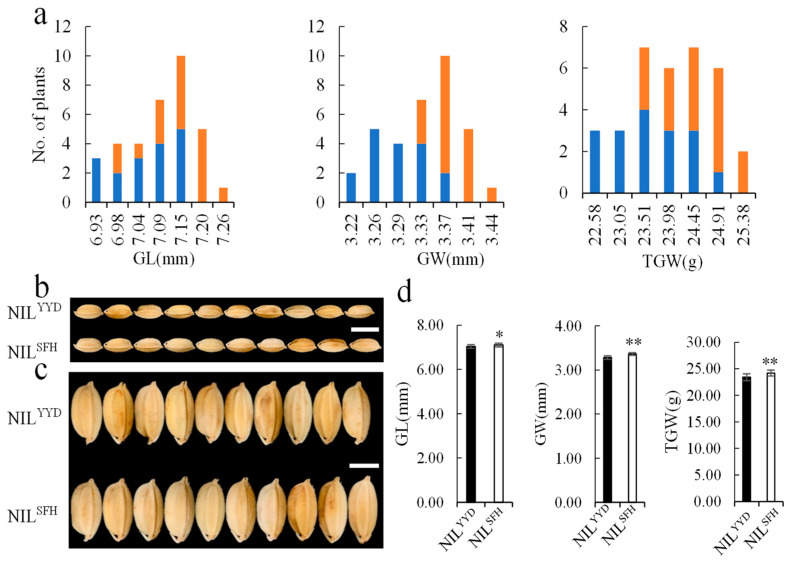
Comparative analysis of grain traits between NIL^YYD^ and NIL^SFH^ for *qGW2-1*. (**a**) Distributions of GL, GW, and TGW in the NIL population. Blue column: the plants carrying the homozygous YYD allele; orange column: the plants carrying the homozygous SFH alleles. (**b**) GL, scale bar: 10 mm. (**c**) GW, scale bar: 5 mm. (**d**) Phenotypes of NIL^YYD^ and NIL^SFH^ in GL, GW, and TGW. * and ** represent significance levels at 0.05 and 0.01, respectively.

**Table 1 plants-13-00770-t001:** QTLs for yield traits detected in the F_2_ population.

Traits	*QTL*	Chr.	Marker Interval	Position	LOD	A	D	R^2^ (%)
GYPP	*qGYPP3*	3	JD3011-YS3041	22.34	5.86	−5.37	1.12	12.35
	*qGYPP6*	6	RM19472-RM19572	30.43	4.43	−4.29	−1.11	8.95
	*qGYPP7*	7	RM22039-JD7015	111.44	4.34	−3.83	−3.11	10.04
NPPP	*qNPPP3*	3	RM14849-RM15029	89.92	7.61	5.5	−4.1	17.85
	*qNPPP4*	4	RM17096-RM17303	176.06	5.83	−3.0	0.3	10.82
	*qNPPP10*	10	RM25367-RM25717	59.62	5.35	−3.3	−0.3	10.76
	*qNPPP12*	12	JD12010-RM27507	11.64	3.76	1.6	2.0	7.58
TNSP	*qTNSP1*	1	JD1016-JD1007	24.55	3.37	−243.3	27.6	6.19
	*qTNSP3*	3	RM15029-RM15217	114.19	3.61	492.0	−462.3	11.30
	*qTNSP7*	7	RM21719-RM21993	80.99	4.18	−300.3	17.1	8.56
	*qTNSP10*	10	RM25367-RM25717	59.70	3.75	−330.9	−170.3	13.15
	*qTNSP12*	12	JD12010-RM27507	11.64	4.05	341.9	−5.8	8.30
NFGPP	*qNFGPP3*	3	JD3011-YS3041	22.34	5.44	−200.6	46.1	11.36
	*qNFGPP6*	6	RM19472-RM19572	30.43	3.86	−146.1	−53.7	7.71
	*qNFGPP7*	7	RM22039-JD7015	111.44	6.19	−179.1	−148.1	16.54
	*qNFGPP12*	12	YS12002-JD12019	50.69	3.20	7.9	218.0	8.16
NSPP	*qNSPP1*	1	RM11570-JD1004	133.83	5.03	−18.8	−15.4	10.40
	*qNSPP3*	3	RM15029-RM15217	114.15	4.47	−26.7	16.3	7.19
	*qNSPP4*	4	RM17096-RM17303	176.00	3.39	16.9	−5.4	6.76
	*qNSPP7*	7	RM22039-JD7015	111.44	11.92	−31.0	−11.4	25.11
SSR	*qSSR1*	1	RM11811-RM3403	174.36	4.55	−8.82	−5.22	8.70
	*qSSR3*	3	JD3011-YS3041	22.32	3.86	−10.07	−1.27	9.16
	*qSSR6*	6	RM19472-RM19572	30.53	5.00	−5.18	−12.31	11.88
	*qSSR10*	10	RM25367-RM25717	59.66	4.00	13.88	−5.66	10.64
	*qSSR12*	12	JD12014-JD12010	0.04	3.35	−0.84	−13.08	8.08
GL	*qGL3*	3	RM3513-RM15981	148.73	9.40	0.50	−0.41	36.01
	*qGL7*	7	RM22039-JD7015	111.46	3.75	0.20	0.18	9.71
	*qGL11*	11	RM3133-RM26567	38.72	3.49	0.28	0.05	12.06
GW	*qGW1*	1	JD1007-RM10609	43.43	3.47	−0.01	0.10	8.26
	*qGW2-1*	2	JD2001-JD2029	70.21	9.02	0.12	0.02	23.58
	*qGW2-2*	2	JD2030-RM1367	99.12	9.85	0.13	−0.01	25.89
	*qGW3*	3	RM16-RM15576	140.87	4.34	0.11	−0.04	10.81
	*qGW10*	10	RM25367-RM25717	59.69	3.89	0.06	0.10	17.81
TGW	*qTGW2*	2	JD2030-RM1367	99.14	4.40	1.62	−0.43	11.51
	*qTGW3*	3	RM15029-RM15353	114.32	8.01	2.68	−0.73	20.67
	*qTGW7*	7	RM22039-JD7015	111.32	3.94	0.97	1.35	10.07

A: additive effect, positive addictive effect means SFH allele increasing trait values; D, dominance effect; R^2^, proportion of phenotype variance explained by the QTL. GYPP, grain yield per plant; NPPP, number of panicles per plant; TNSP, total number of spikelets per plant; NFGPP, number of filled grains per plant; NSPP, number of spikelets per panicle; SSR, seed-setting rate; GL, grain length; GW, grain width and TGW, 1000-grain weight.

**Table 2 plants-13-00770-t002:** QTLs of grain size in three F_7_ populations.

Population	Trait	Interval	Sample	LOD	A	D	R^2^ (%)
YM1	GL	W236-YS2006	128	ns	ns	ns	ns
	GW	W236-YS2006	128	8.14	0.09	−0.01	31.54
	TGW	W236-YS2006	128	4.10	1.21	−0.55	19.07
YM2	GL	YS2024-JD2029	140	ns	ns	ns	ns
	GW	YS2024-JD2029	140	13.07	0.05	−0.01	45.79
	TGW	YS2024-JD2029	140	3.56	0.57	−0.04	14.70
YM3	GL	W236-YS2010	179	ns	ns	ns	ns
	GW	W236-YS2010	179	3.33	0.06	0.01	11.17
	TGW	W236-YS2010	179	4.24	0.97	−0.18	17.77

A: additive effect, positive addictive effect means SFH allele increasing trait values; D, dominance effect; R^2^, proportion of phenotype variance explained by the QTL. ns, no significance.

## Data Availability

Data are contained within the article.

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
