# Peer review of "Dissecting the Genetic Basis of Yield Traits and Validation of a Novel Quantitative Trait Locus for Grain Width and Weight in Rice"

_plants, 2024, doi:10.3390/plants13060770_

Round 1
Reviewer 1 Report
Comments and Suggestions for Authors
1. Spell out QTL for the first time in abstract
2. Provide one line conclusion in abstract
3. Objective and hypothesis are missing in the introduction part. Please provide
4. Why the yield of SFH is too low. Any reason?
5. Divide the discussion into subheadings. Discussion needs improvement.
6. Did the authors check the normality of the data? In such smaller datasets, it is hard to get normality and then one cannot perform ANOVA.
Comments on the Quality of English Language
Need improvement.
Author Response
Reviewer 1
1. Spell out QTL for the first time in abstract
Response: Thank you very much for your careful review. We have spelled QTL for the first time in abstract.
- Provide one line conclusion in abstract
Response: We have provided one line conclusion in abstract (Lines 24 to 27).
- Objective and hypothesis are missing in the introduction part. Please provide
Response: We have provided objective and hypothesis in the introduction part (Lines 31 to 47).
- Why the yield of SFH is too low. Any reason?
Response: Rice yield per plant is mainly determined by three key components including number of panicles per plant, number of filled grains per panicle and grain weight. SFH has only a small number of effective panicles (3-4 panicles per plant) and very few grains per panicle, which leads to its low yield.
- Divide the discussion into subheadings. Discussion needs improvement.
Response: We have divided the discussion into four subheadings and conducted further discussions (Lines 362 to 368, Lines 399 to 403).
- Did the authors check the normality of the data? In such smaller datasets, it is hard to get normality and then one cannot perform ANOVA.
Response: Thank you for your suggestion. Sorry for failing to conduct normality analysis on the data, so we have deleted the NIL population data for ANOVA analysis (Table 3.).
Reviewer 2 Report
Comments and Suggestions for Authors
Dissecting the genetic basis of yield traits and validation of a 2 novel quantitative trait locus for grain width and weight in rice
Rice is one of the most important cereal crops and serves as a stable food for more than half the world population. For enhancing rice production, more attention is directed to deeply explore the genetic mechanism of rice yield. This is very good concept for growing population, decreasing cultivated land, natural disasters and increasing demand make threats against food security.
This paper explores on the correlations among 9 yield traits were analyzed. The grain yield per plant positively correlated with the six yield traits except grain length and grain width and showed the highest correlation coefficient of 0.98 with the number of filled grains per plant. A genetic map containing 133 DNA 19 markers was constructed and spanned 1831.7 cM throughout 12 chromosomes. A total of 36 QTLs for yield traits were detected on 9 chromosomes except for the remaining chromosomes 5, 8 and 9. The phenotypic variation explained by a single QTL ranged from 6.19% to 36.01%. Furthermore, a major QTL for grain width and weight, qGW2-1, was confirmed to be newly identified and narrowed down to a relatively smaller interval about ~2.94-Mb. The authors established map-based cloning and molecular design breeding in rice.
However, I have few concerns on current manuscript:
1. References are missing in materials and methods section mostly. Should be updated all references.
2. The correlation coefficients among nine traits in F2 population of table1 will be represents as a correlation coefficient plot. There are many visualization programs like R program packages.
3. In this study authors identified 36 QTLs for 9 yield traits. However, what are the major key contribution of these QTLs in map-based cloning and molecular design breeding in rice.
4. Did authors identify any novel QTLs for enhancing rice production and genetic mechanism of rice yield?
5. If these identified QTLs are integrate with single cell RNA sequence of paddy plants it may very strong and support to author’s concept but it’s just suggestion.
This manuscript is good in shape and well-established objectives of this study. I’m happy to accept this manuscript after minor corrections.
Author Response
Rice is one of the most important cereal crops and serves as a stable food for more than half the world population. For enhancing rice production, more attention is directed to deeply explore the genetic mechanism of rice yield. This is very good concept for growing population, decreasing cultivated land, natural disasters and increasing demand make threats against food security.This paper explores on the correlations among 9 yield traits were analyzed. The grain yield per plant positively correlated with the six yield traits except grain length and grain width and showed the highest correlation coefficient of 0.98 with the number of filled grains per plant. A genetic map containing 133 DNA 19 markers was constructed and spanned 1831.7 cM throughout 12 chromosomes. A total of 36 QTLs for yield traits were detected on 9 chromosomes except for the remaining chromosomes 5, 8 and 9. The phenotypic variation explained by a single QTL ranged from 6.19% to 36.01%. Furthermore, a major QTL for grain width and weight, qGW2-1, was confirmed to be newly identified and narrowed down to a relatively smaller interval about ~2.94-Mb. The authors established map-based cloning and molecular design breeding in rice.However, I have few concerns on current manuscript:
- References are missing in materials and methods section mostly. Should be updated all references.
Response: Thank you very much for your careful review. We have supplemented the reference materials for the Materials and Methods section (Lines 441, 455).
- The correlation coefficients among nine traits in F2 population of table1 will be represents as a correlation coefficient plot. There are many visualization programs like R program packages.
Response: We have represented the correlation coefficients among 9 yield traits in the F2 population as a correlation coefficient plot in figure 4.
- In this study authors identified 36 QTLs for 9 yield traits. However, what are the major key contribution of these QTLs in map-based cloning and molecular design breeding in rice.
Response: Thirty six QTLs detected in the primary mapping show the genetic effects and provide primary locations for further map-based cloning and molecular design breeding in rice. For example, qGW2-1, which contributes significantly to the variation of grain width and weight, was detected in the initial localization on chromosome 2. After validation of genetic effects for qGW2-1 using genetic populations, map-based cloning can be performed.
- Did authors identify any novel QTLs for enhancing rice production and genetic mechanism of rice yield?
Response: After compared to the QTL locations, no QTL for GW and TGW showed the overlapping interval with qGW2-1. Therefore, qGW2-1 that was detected in this study is a novel QTL for rice yield.
- If these identified QTLs are integrate with single cell RNA sequence of paddy plants it may very strong and support to author’s concept but it’s just suggestion.
Response: Thank you very much for your suggestion. Single cell RNA sequencing technology can achieve rapid sequencing and analysis of thousands of cells, detecting differentially expressed genes between cell populations. Combining single-cell RNA sequencing technology with QTL detection is more conducive to its rapid localization. In future research, we will consider combining single-cell RNA sequencing with QTL mapping.
Reviewer 3 Report
Comments and Suggestions for Authors
This manuscript by Yin et al. provides useful new information on rice yield related traits. So, it can be published after minor revisions.
Point 1: Clearly articulate the motivation behind studying rice yield traits and the significance of addressing food security concerns. Provide more recent references to support the context of the challenges faced in rice production.
Point 2: Define acronyms such as YYD/SFH and GWAS for readers who may not be familiar with these terms.
Point 3: Strengthen the link between tillering ability and its impact on NPPP. Elaborate on the significance of MOC1, MOC2, and MOC3 genes and their role in tillering.
Point 4: Table 1 must be presented in figure form.
Point 4: Check references they are inconsistent. Journal names must be according to Plants journal format.
Comments on the Quality of English LanguageCan be improved
Author Response
This manuscript is good in shape and well-established objectives of this study. I’m happy to accept this manuscript after minor corrections. This manuscript by Yin et al. provides useful new information on rice yield related traits. So, it can be published after minor revisions.
- Clearly articulate the motivation behind studying rice yield traits and the significance of addressing food security concerns. Provide more recent references to support the context of the challenges faced in rice production.
Response: Thank you very much for your careful review. We have added some materials to articulate the motivation behind studying rice yield traits and the significance of addressing food security concerns, and provided recent reference materials to support the challenges faced by rice production (Lines 31-47).
- Define acronyms such as YYD/SFH and GWAS for readers who may not be familiar with these terms.
Response: We have defined acronyms YYD/SFH and GWAS (Lines 16, 55).
- Strengthen the link between tillering ability and its impact on NPPP. Elaborate on the significance of MOC1, MOC2, and MOC3 genes and their role in tillering.
Response: We have added some materials to strengthen the link between tillering ability and its impact on NPPP, and elucidate the significance of MOC1, MOC2, and MOC3 genes and their roles in tillering (Lines 67, 73-83).
- Table 1 must be presented in figure form.
Response: We have represented the correlation coefficients between 9 personality traits in the F2 population as a correlation coefficient plot in figure 4.
- Check references they are inconsistent. Journal names must be according to Plants journal format.
Response: We have checked the reference materials and ensured that the journal name conforms to the Plants journal format.
Round 2
Reviewer 1 Report
Comments and Suggestions for Authors
The authors incorporated all comments.